# Fluorescence In Situ Hybridization (FISH) Tests for Identifying Protozoan and Bacterial Pathogens in Infectious Diseases

**DOI:** 10.3390/diagnostics12051286

**Published:** 2022-05-21

**Authors:** Jyotsna S. Shah, Ranjan Ramasamy

**Affiliations:** IGeneX Inc. and ID-FISH Technology Inc., 556 Gibraltar Drive, Milpitas, CA 95035, USA

**Keywords:** diagnostic tests, FISH tests, fluorescence in situ hybridization, *Babesia duncani*, *Babesia microti*, LED fluorescence microscopy, *Mycobacterium avium*, *Mycobacterium tuberculosis*, pathogen identification, *Plasmodium falciparum*, *Plasmodium knowlesi*, *Plasmodium vivax*, ribosomal RNA

## Abstract

Diagnosing and treating many infectious diseases depends on correctly identifying the causative pathogen. Characterization of pathogen-specific nucleic acid sequences by PCR is the most sensitive and specific method available for this purpose, although it is restricted to laboratories that have the necessary infrastructure and finance. Microscopy, rapid immunochromatographic tests for antigens, and immunoassays for detecting pathogen-specific antibodies are alternative and useful diagnostic methods with different advantages and disadvantages. Detection of ribosomal RNA molecules in the cytoplasm of bacterial and protozoan pathogens by fluorescence in-situ hybridization (FISH) using sequence-specific fluorescently labelled DNA probes, is cheaper than PCR and requires minimal equipment and infrastructure. A LED light source attached to most laboratory light microscopes can be used in place of a fluorescence microscope with a UV lamp for FISH. A FISH test hybridization can be completed in 30 min at 37 °C and the whole test in less than two hours. FISH tests can therefore be rapidly performed in both well-equipped and poorly-resourced laboratories. Highly sensitive and specific FISH tests for identifying many bacterial and protozoan pathogens that cause disease in humans, livestock and pets are reviewed, with particular reference to parasites causing malaria and babesiosis, and mycobacteria responsible for tuberculosis.

## 1. Background

The in-situ hybridization (ISH) technique for examining the formation and detection of RNA-DNA or DNA-DNA nucleotide complementary hybrids in cells utilizing radioactively labelled oligonucleotides as probes was first described in 1969 [1,2]. ISH permits the detection of nucleic acids in individual cells that contain specific nucleotide sequences among a heterogenous population of cells. It also allows the simultaneous determination of biochemical and morphological characteristics of the reactive cells. The hybridization of fluorescently labeled, chromosome-specific, composite DNA probe pools to cytological preparations, termed chromosome painting, has made major contributions to karyotyping and identifying chromosomal changes responsible for human pathology [3]. Fluorescence in situ hybridization (FISH) methods have since been developed to study chromosomal genomic changes at the kilobase level [4]. ISH was first utilized for bacteriology in 1983 with radioactively labeled DNA probes targeting ribosomal RNA (rRNA) [5]. Fluorescently labeled probes have subsequently replaced radioactive probes for FISH [6,7,8]. In 1989, DeLong and colleagues demonstrated that oligodeoxynucleotide probes, complementary to 16S rRNA, labelled with different fluorescent molecules used in FISH can detect single microbial cells and differentiate closely related organisms [9,10]. Shah et al. in 1990 established that FISH can detect and differentiate *Pneumocystis carinii* strains in sputum and tissue from patients [11]. FISH assays for detecting pathogens in clinical samples now use either peptide nucleic acid (PNA) probes (in which the sugar phosphate backbone is replaced with a more hydrolysis-resistant polyamide chain), locked nucleic acid (LNA) probes (where greater stability is achieved by a methylene bridge linking the 2′ oxygen to the 4′ carbon of the pentose) or, more commonly, DNA probes [12,13,14,15,16,17,18,19,20].

The application of FISH assays for detecting and identifying microbial pathogens has advanced considerably since the turn of the century [21]. FISH techniques have been applied to investigate the localization of viral nucleic acids within infected tissues and organs, e.g., for SARS-CoV-2 [22,23] and HIV [24], and rarely for identifying the infecting virus [25]. PCR tests, and the detection of viral antigens and specific antibodies to viral antigens, are more commonly and effectively used for diagnosing viral infections. In contrast, FISH tests have proved useful for identifying bacterial, fungal and protozoan disease-causing pathogens, particularly at the species level. Recent examples are listed in Table 1.

Ribosomal RNA molecules possess genus, species and strain-specific regions. Hence, 16S rRNA sequences have been used to establish phylogenetic relationships among bacteria [47]. The binding of rRNA-targeting probes can be visualized without nucleic acid-based amplification (NAA) of the target rRNA sequence because rRNA is present in each of the numerous ribosomes in the cytoplasm. Many FISH tests listed in Table 1 are based on DNA or PNA probes that hybridize to specific rRNA sequences in suitably permeabilized cells. Due to the complex three-dimensional structure of rRNA, not all nucleotide sequences within an rRNA molecule are equally accessible for hybridizing with FISH probes. Loop and hairpin formation as well as rRNA-protein interactions hinder hybridization and produce differential binding sensitivity with oligonucleotide probes [48,49]. It is therefore necessary to evaluate and optimize every newly designed probe with the respective reference organism and appropriate negative controls, before it is applied to test samples. Self-annealing and hairpin formation occurring within an oligonucleotide probe itself can lead to low signal intensities, and hence newly designed oligonucleotides also need to be checked for internal complementarity using appropriate software.

The World Health Organization estimated 241 million cases of malaria and 627,000 resulting deaths worldwide in 2020 [50]. Infection with *Mycobacterium tuberculosis* (MTB), which is primarily responsible for human tuberculosis, caused an estimated 1.4 million deaths worldwide in 2020 [51]. Malaria and tuberculosis can manifest as latent infections that rapidly become fulminant diseases with fatal consequences. Babesiosis is a potentially fatal, tick-borne, globally emerging human disease, that also afflicts livestock and domestic pets [52]. While the principle of FISH tests remains the same, their application to identify different pathogens can vary considerably. Recently developed FISH tests that can be easily used in resource-limited laboratories worldwide for identifying causative pathogens in malaria, tuberculosis and babesiosis are therefore selected for detailed consideration in this article. These FISH tests for malaria, tuberculosis and babesiosis have the following shared characteristics: (i) the assays are performed on thin smears on glass microscope slides, (ii) cells in the smear are rapidly permeabilized for hybridization which is then performed with fluorescently labelled DNA probes at 37 °C for 30 min, (iii) fluorescence can be viewed under LED illumination in common light microscopes as shown in Appendix A, (iv) the test is completed in less than two hours, and (v) only living cells are labelled because rRNA is rapidly degraded in dying cells. Appendix A summarizes the published work–flow for tuberculosis FISH tests [28,29]. 

## 2. FISH Tests for Malaria

### 2.1. Background

*Plasmodium falciparum* is responsible for most of the annual 241 million global malaria infections, together with an estimated 4.5 million cases of *Plasmodium vivax* and fewer cases of *Plasmodium malariae and Plasmodium ovale* [50]. *Plasmodium knowlesi**,* which normally infects macaque and leaf monkeys, causes a significant number of dead-end human infections in Southeast Asian countries [53]. *Plasmodium knowlesi* is difficult to differentiate from human malaria parasites in Giemsa-stained blood smears that are commonly used for diagnosing malaria in endemic areas [53]. Occasional zoonotic infections with *Plasmodium cynomolgi* and *Plasmodium inui* have also been reported in Southeast Asia [53]. *Plasmodium brasilianum* and *Plasmodium simium* infect platyrrhine monkeys in South and Central America, are genetically almost identical to the human malaria parasites *P. malariae* and *P. vivax* respectively, and likely to have been derived from the human parasites by anthroponosis [53]. *P. simium* and *P. brasilianum* can also infect humans by zoonosis but their differentiation from *P. vivax* and *P. malariae* respectively in stained blood smears is not possible [53]. 

There were 2171 US cases of malaria reported to the US Centers for Disease Control and Prevention (CDC) in 2017 according to the latest available CDC report [54]. *Plasmodium falciparum* accounted for 70.5%, *P. vivax* 10.0%, *P. ovale* 5.5%, and *P. malariae* 2.6% of the infections, all of which had been acquired outside the US [54]. Infections with two or more *Plasmodium* species were responsible for 1.0% of infections [54]. The identification of malaria parasites for diagnosis is therefore also needed in the US and other countries with no indigenous transmission of malaria. 

Giemsa-stained thick and thin blood smear microscopy has been the most widely used technique globally for diagnosing malaria. It however requires time and an experienced microscopist for optimal sensitivity of detection and for identifying the infecting species of parasite. A sensitivity of >150 parasites per µL is typically achieved during routine microscopy [39,55]. Rapid diagnostic tests (RDTs), based on the immunochromatographic detection of antibodies to the histidine rich protein 2 of *P. falciparum* (PfHRP2) and pan *Plasmodium*-specific lactate dehydrogenase and aldolase, have more recently proved helpful in resource-limited locations [55]. However, selection for *PfHRP2* gene deletions in *P. falciparum* in malaria-endemic areas of Africa has lately increased the false negativity rates for *P. falciparum* [55]. PCR-dependent NAA diagnostic tests have the best sensitivity and specificity and are able to identify *Plasmodium* at the species level, but the procedure is not suitable for resource-poor settings and field use [56]. The Loop-Mediated Isothermal Amplification (LAMP) technique has the desired sensitivity and specificity but is not widely utilized for routine malaria diagnosis and species identification [57]. Flow cytometric detection of malaria parasites in blood have also been recently described although details of its limit of detection and ability to identify different species remain to be established [58,59]. 

### 2.2. Genus-Specific FISH Test That Identifies All Common Human Malaria Parasites 

Simple, rapid and specific FISH tests for malaria, that can be easily performed in resource-constrained diagnostic laboratories, have many advantages [39,42]. These FISH tests for malaria employ a similar protocol to FISH tests for tuberculosis shown in Appendix A. A standard laboratory microscope with a LED fluorescence unit attached to it can be used to read the processed smears on the slide (Appendix A). Results from a *Plasmodium* genus-specific FISH test utilizing a DNA probe hybridizing to 18S rRNA [39] are reproduced in Figure 1. 

The *Plasmodium* genus-specific FISH test identified all common species of human malaria parasites with 100% specificity when compared with several other common human blood-borne pathogens [39]. 

### 2.3. FISH Test for Specifically Identifying Plasmodium Falciparum

A PF-FISH test that complements the *Plasmodium* genus-specific FISH test, and designed to specifically identify *P. falciparum*, utilized a mixture of *P. falciparum* 18S rRNA-specific probes labeled with Alexa 488 green and the *Plasmodium* genus-specific probe labeled with a Texas Red in a multiplex format. *Plasmodium falciparum* fluoresced green (Figure 2), while all *Plasmodium* parasites, including *P. falciparum,* fluoresced red with appropriate light filters in the PF-FISH test [39].

### 2.4. FISH Test for Specifically Identifying Plasmodium Vivax

In a second complementary FISH test, termed the PV-FISH test, a mixture of DNA probes that hybridize only to the 18S rRNA of *P. vivax* were labeled with the Alexa 488 green, and used in a multiplex format with the *Plasmodium* genus-specific probe labeled with Texas Red. Only *P. vivax* fluoresced green, and all *Plasmodium* species fluoresced red, with appropriate filters in the PV-FISH test [39] as shown in Figure 3. 

The two FISH tests for specifically identifying *P. falciparum* and *P. vivax* had greater analytical sensitivity, and also higher clinical sensitivity and specificity, compared to microscopic examination of Giemsa-stained blood smears [39].

### 2.5. FISH Test for Specifically Identifying Plasmodium Knowlesi

Zoonotic *P. knowlesi* infections in Southeast Asia are commonly misidentified as *P. malariae* or *P. falciparum* in Giemsa-stained human blood smears because of morphological similarities between the blood stages, so that PCR-based tests were needed for confirming *P. knowlesi* infections [60,61,62,63,64,65]. Correct diagnosis of *P. knowlesi* malaria is essential for two reasons: (i) understanding its epidemiology, and (ii) its pathogenicity and drug treatment options can differ from human malaria caused by *P. falciparum*, *P. malariae*, *P. ovale* and *P. vivax* that also occur in Southeast Asia. Reliable RDTs for specifically detecting *P. knowlesi* are not yet available [66]. However, a simple and specific FISH test using DNA probes targeting *P. knowlesi* 18S rRNA (termed the PK-FISH test) specifically identified *P. knowlesi* in blood smears [42], as shown in Figure 4.

The PK-FISH test, like the analogous *Plasmodium* genus-specific test and the PF-FISH and PV-FISH tests [39], identified all asexual blood stages, i.e., rings, trophozoites and schizonts, as shown in Figure 5 for *P. knowlesi* [42]. The PK-FISH test also detected *P. knowlesi* at the low limit of 16 *P. knowlesi* parasites per µL, even in the concomitant presence of *P. falciparum* at approximately 500 parasites per µL [42], which is superior to that possible with routine microscopic examination of Giemsa-stained thin blood films [39,55]. This property is very useful in Southeast Asia where mixed infections of *P. knowlesi* and other human malaria parasite species are common [60,63]. The highly specific and sensitive PK-FISH therefore meets a widely—recognized diagnostic need of peripheral and district-level clinical laboratories in areas of Southeast Asia where *P. knowlesi* zoonosis is prevalent [53,60,61,62,63,64,65,66].

### 2.6. Conclusions

A large number of malaria tests are performed for diagnostic and screening purposes in malaria-endemic countries. Tests in malaria-free countries are utilized for (i) screening passengers arriving from malaria-endemic countries to prevent the reintroduction of malaria if mosquito vectors are present in the country of arrival, and (ii) confirming malaria in arriving travelers who have malaria-like symptoms. FISH tests are more costly and complex to perform than Giemsa-stained blood smear microscopy and RDTs, but significantly less so than NAA-dependent PCR and LAMP, for detecting *Plasmodium* infections (table in Section 5 below). FISH tests are particularly useful for identifying the species of infecting parasites, as illustrated here for *P. falciparum*, *P. vivax* and *P. knowlesi*. Therefore, the clinical diagnostic characteristics and simple methodology of the newly described FISH tests for malaria parasites, suggest that they can (i) usefully complement Giemsa-stained blood smear microscopy and RDTs for routine diagnosis and screening for malaria, and (ii) identify the species of infecting *Plasmodium* (including in mixed infections), in both endemic and non-endemic countries.

## 3. FISH Tests for Babesiosis

### 3.1. Background

*Babesia* are apicomplexan protozoan parasites, like *Plasmodium*, that infect and replicate within red blood cells to cause babesiosis in humans, livestock and pets [52,67,68,69]. Ticks acquiring *Babesia* from animal reservoirs function as vectors to transmit infections to humans [52,67,68,69]. Infections can also be transmitted congenitally and through blood transfusion [69,70]. The CDC recorded 2418 cases of babesiosis in 2019 in the 40 US states and the District of Columbia where babesiosis was a notifiable disease [70]. *Babesia microti*, *B. duncani* and *B. divergens* are largely responsible for human infections in the US [67,68,69,70]. *Babesia microti*, *B. divergens*, *B. venatorum* and *B. crassa* are responsible for human babesiosis in Eurasia [52,68,69,70]. Human babesiosis is also prevalent in Africa, South America and Australia [68,69]. Moreover, *Babesia* infect cattle (e.g., *B. bovis*, *B. divergens*, *B. bigemina*), horses (e.g., *B. caballi*), dogs (e.g., *B. canis*), cats (*B. felis*), deer (e.g., *B. odocolei*, *B.venatorum*), mice (e.g., *B. microti*, *B. rodhani*) and other animals [52,69]. The hard ticks, *Ixodes scapularis*, *I. ricinus*, *I. persulcatus* and *Dermacentor albipictus* are vectors that transmit *Babesia* to humans [69,70,71]. 

Laboratory tests commonly used for diagnosing babesiosis involve the detection of (i) parasites in stained blood smears by microscopy, (ii) serum antibodies to *Babesia* by immunoassays, and (iii) *Babesia*-specific nucleic acid sequences by PCR [69,70]. Babesiosis and borreliosis (a tick vector-borne disease caused by spirochete *Borrelia* bacteria), share many clinical manifestations, and occur as coinfections [72,73,74,75,76]. They have an overlapping geographical distribution [67,68,69,70,71,77,78], underscoring the importance of diagnostic laboratory tests for differentiating babesiosis and borreliosis. Early intra-erythrocytic stages of human—infecting *Babesia* species are not readily distinguished from the ring and trophozoite stages of *P. falciparum* by microscopy in areas where babesiosis and malaria are co-endemic [69]. Furthermore, the antibody assays for diagnosing human and veterinary babesiosis cannot easily differentiate between active and resolved *Babesia* infections [67,68,69,70]. PCR tests for babesiosis have high sensitivity [79,80,81] and are recommended for screening donor blood for babesiosis in the US [82]. However, cost and infrastructure requirements make PCR-based tests impractical for use in resource-limited laboratories and field settings. 

### 3.2. Babesia Genus-Specific FISH Test

A FISH test that identifies all common species of *Babesia* parasites, with many advantages for use in resource-limited laboratories, has been developed [43,44]. Termed the *Babesia* genus FISH test, it is based on DNA probes that specifically hybridize to the multiple copies of *Babesia* 18S rRNA present in the parasite cytoplasm. Like other rRNA-directed FISH tests, the *Babesia* genus FISH test does not require NAA—a process that is sensitive to NAA inhibitors sometimes present in blood [83].

The *Babesia* genus-specific FISH test detects *B. microti, B. duncani* and *B. divergens*, as well as the two important parasites causing bovine babesiosis, *B. bovis* and *B. bigemina* [43], as illustrated in Figure 6.

The *Babesia* genus-specific FISH test, in conjunction with an IFA test for detecting serum antibodies to *B. duncani* and *B. microti,* on clinical samples originating from USA, Australia, Europe and elsewhere, showed that the global prevalence of *B. duncani* infections had hitherto been under-estimated [44]. Furthermore, the *Babesia* genus-specific FISH test was highly specific and did not detect other pertinent pathogens found in human blood [43], including different species of *Borrelia* and *Plasmodium* [43], as well as various species of *Bartonella* that infect humans and domestic pets [33,84]. 

### 3.3. Conclusions

The prevalence of human babesiosis has probably been underestimated throughout the world [44]. Babesiosis also afflicts livestock and pets [52]. The clinical diagnostic characteristics and simple methodology of the FISH tests show that they can complement existing diagnostic methods to meet an increasing need to specifically and easily identify *Babesia* infections in patients and animals. FISH tests can be used in mixed infections. FISH is also useful for histopathological investigations to identify *Babesia* parasites sequestered in tissues [45]. Species-specific *Babesia* FISH tests, that are presently being developed, can address more precise diagnostic requirements in babesiosis. 

## 4. FISH Tests for Tuberculosis

### 4.1. Background

Pulmonary mycobacterial infections in humans are caused mostly by *Mycobacterium tuberculosis* (MTB) and the closely related species *Mycobacterium bovis*, both of which belong to the *Mycobacterium tuberculosis* complex (MTBC) [51]. Infections with non-tuberculous mycobacteria (NTM), including the *Mycobacterium avium* complex (MAC), *M. kansasii, M. fortuitum, M. xenopi, M. abscessus,* and *M. simiae* also occur worldwide, making their differential diagnosis important for clinical purposes [51,85,86,87]. Infections with MAC are common in late-stage human immunodeficiency virus infections, where the mycobacteria are often restricted to lymphoid tissue. FISH provides a sensitive and specific method for detecting MAC by in biopsied tissues, which is important because the management and treatment of patients with MTBC and NTM infections are different [30]. Norcardiosis, caused by related *Norcadia* species widely distributed in the environment, also needs to be differentiated from MTB during lung infections [88].

Microscopic examination for acid-fast staining (AFS) bacilli, e.g., with the Ziehl-Neelsen stain, in sputum or tissue plays an important role in the diagnosis of tuberculosis [89]. AFS does not differentiate between mycobacterial species. It also lacks sufficient sensitivity with sputum smears and tissue samples. Sensitivity is increased in sputum smears and biopsied tissue by staining with auramine and detecting fluorescence in a LED fluorescence microscope [90], similar to that used for FISH tests (Appendix A). The Xpert^®^ MTB/RIF system or Xpert (Cepheid, Sunnyvale, CA, USA), is a PCR-based nucleic acid amplification (NAA) technique that detects specific DNA sequences of MTB [91,92,93]. Xpert is recommended by the WHO for identifying MTB and rifampicin resistance in the sputum of adults and children presumed to have tuberculosis [91,92,93]. It is approximately 100 times more sensitive for detecting MTB than conventional AFS, but Xpert identifies only MTB, and its use in many resource-constrained endemic countries is limited by cost. Culturing clinical specimens continues to have an important role in identifying infecting mycobacteria in tuberculosis-like disease, especially in smear negative, pediatric or extra pulmonary infections and resource-limited laboratories. Culture techniques that significantly reduce culture times for identifying mycobacteria are becoming available to facilitate diagnosis [94]. Immunochromatographic tests to detect specific proteins produced by MTBC are more cost-effective than PCR tests, but, as yet, do not have the desired specificity and sensitivity [92,93]. 

### 4.2. FISH Tests for Identifying the Genus Mycobacterium as Well as the Mycobacterium Tuberculosis and Mycobacterium Avium Complexes in Culture

A simple and rapid test for directly identifying MTB and NTM in sputum and tissues that can be used by resource-limited laboratories in endemic countries is therefore expected to greatly aid tuberculosis control worldwide [92,93]. Two dual color FISH tests, with simple protocols (Appendix A), and requiring only a LED fluorescence microscope (Appendix A), meet this need [28,29,30]. The MN Genus-MTBC FISH test used an orange fluorescent DNA probe that specifically hybridizes to the 23S rRNA of the *Mycobacterium tuberculosis* complex (MTBC) and a green fluorescent probe specific for the *Mycobacterium* and *Nocardia* genera (MN Genus) 16S rRNA to detect and distinguish MTBC from other mycobacteria and *Nocardia* species. A complementary MTBC-MAC FISH test used green and orange fluorescent probes for 23S rRNA that respectively differentiate MTBC and MAC [28,29,30].

All *Mycobacterium* species from reference cultures, except *M. wolinskyi*, reacted positively with the MN Genus-specific probe and only the *M. tuberculosis* complex species reacted positively with the MTBC- specific probe in the MN Genus—MTBC FISH test. Only the *M. tuberculosis* complex species reacted positively with the MTBC-specific probe and only the *M. avium* complex species reacted positively with the MAC–specific probe in the MTBC-MAC FISH test [28]. *Nocardia* reacted positively with the MN Genus probe but not with the MTBC- and MAC-specific probes in the MN Genus-MTBC and the MTBC-MAC tests [28]. The estimated specificity of the two FISH tests for MTBC and MAC in reference cultures was 100%, with a limit of detection of 1.5–5.1 × 10^4^ bacteria per ml [28]. Results from the two FISH tests with reference strain cultures of *M. tuberculosis*, *M. avium* and *M. kansasii* [28] are reproduced in Figure 7.

### 4.3. FISH Tests for Identifying MTBC and MAC in Sputum 

The FISH tests used for culture identification can also be used for directly detecting mycobacteria in sputum [29]. Figure 8 reproduces results obtained with the MN Genus-MTBC FISH test performed directly on a sputum smear containing MTBC which reacted with the MN Genus- and MTBC-specific probes and a different smear containing *M. abscessus,* an NTM, that reacted only with the MN Genus-specific probe.

### 4.4. Other FISH Tests for Tuberculosis

Another FISH test specific for MTBC in sputum targeting the *rpoB* gene coding for the β subunit of RNA polymerase has been described [95]. The *rpoB* FISH test however required enzyme digestion of sputum, concentration of mycobacteria by centrifugation, hybridization overnight and a UV fluorescence microscope for visualizing results [95]. Other PNA or DNA probe-based FISH tests described for MTBC and MAC [12,13,14,15,16] also require long and more stringent hybridization procedures, and a UV fluorescence microscope for viewing test results. They have not been used yet for routine diagnosis in endemic countries. The MN Genus-MTBC and MTBC-MAC FISH tests on the other hand, cost < US$5 per test, provide results in <2 h after sputum collection, do not require enzyme treatment and centrifugation, can use LED fluorescence microscopy (Appendix A), and utilize reagents that are stable at ambient temperature [29]. They are used in India [30]. All FISH assays have the advantage that they are unaffected by inhibitors in respiratory samples which reduce sensitivity and require elaborate controls for NAA tests [96,97].

### 4.5. Conclusions

FISH tests for tuberculosis meet internationally-expressed needs for diagnosing tuberculosis in respiratory samples [92,93]. They can complement AFS microscopy and NAA methods to detect and differentiate MTBC from MAC and other NTM in sputum and cultures. FISH is also useful in detecting MTB and MAC in biopsied tissues [30]. Table in Section 5 below compares NAA and FISH tests for tuberculosis. 

## 5. Comparison of NAA and FISH Tests for Diagnosing Malaria and Tuberculosis

Tests that depend on amplifying specific nucleic acid sequences of pathogens and the subsequent detection and/or sequencing of the amplified nucleic acids are widely regarded as the gold standard for diagnostic tests because of their high sensitivity and specificity. Two NAA techniques that can be used for RNA and DNA, are based on PCR [98] and LAMP [99]. Diagnostic test needs vary considerably for different pathogens and the diseases caused by them. Malaria and tuberculosis are parasitic and bacterial diseases respectively of great global clinical concern [50,51]. The use of PCR and LAMP tests for identifying pathogens causing the two diseases are therefore compared with FISH tests in Table 2.

## 6. Overall Conclusions and Future Prospects

The simplicity, cost, modest infrastructure/equipment/reagent requirements, reagent stability, good diagnostic parameters, and the ability to identify pathogens at the species level, suggest that FISH tests can be used in advanced as well as resource-constrained diagnostic laboratories throughout the world. FISH tests, therefore, can complement existing diagnostic tests in both disease-endemic and non-endemic countries. FISH tests are particularly advantageous for identifying pathogens at the species level. Flow cytometry combined with FISH has been shown able to rapidly identify potentially pathogenic bacteria present in food, water, air and biofilms formed on various abiotic surfaces [110]. The use of the Flow-FISH methodology [24,110] for identifying causative pathogens in infectious diseases therefore merits further investigation. FISH may also be usefully interfaced with advanced optical and microscopic techniques [22,23,25,111,112] to further expand its scope for identifying infecting pathogens for research and diagnostic purposes.

## Figures and Tables

**Figure 1 diagnostics-12-01286-f001:**
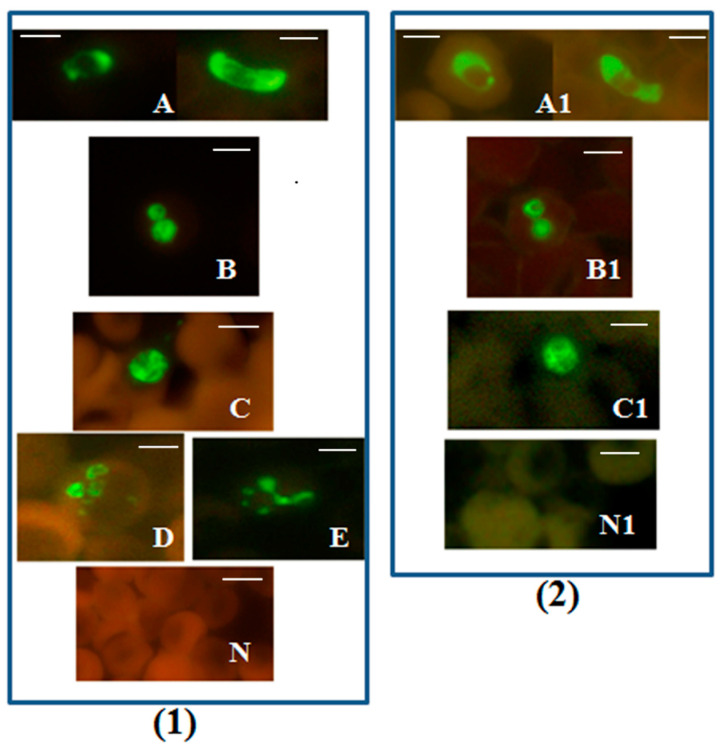
*Plasmodium* genus-specific FISH test identifying all human malaria parasites. Photographs showing *Plasmodium* genus-specific FISH test results with blood smears from patients with confirmed *P. falciparum, P. vivax, P. malariae, P. ovale* and *P. knowlesi* infections. Fluorescence was viewed in (**1**)—fluorescence microscope with a UV light source, and (**2**)—microscope with LED light source illustrated in Appendix A. Green fluorescence demonstrated the presence of *Plasmodium* rRNA. (**A**,**A1**) *P. falciparum* including a crescent shaped gametocyte; (**B**,**B1**) *P. vivax;* (**C**,**C1**) *P. knowlesi*; (**D**) *P. ovale*; (**E**) *P. malariae*; and (**N**,**N1**) negative controls. Alexa 488 green was used to label the *Plasmodium* genus-specific probe in the FISH test. Only *Plasmodium* parasites fluoresce green in the assay. Scale bars represent approximately 5 µm. Figure reproduced with permission under the creative commons license from Reference [39].

**Figure 2 diagnostics-12-01286-f002:**
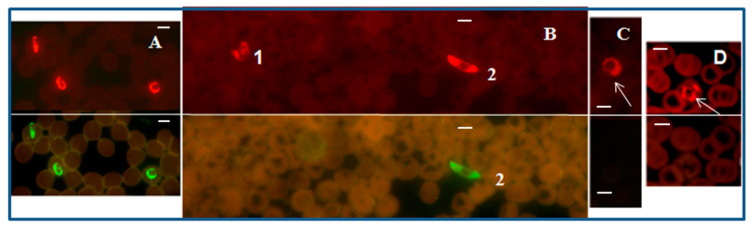
*P. falciparum*-specific PF-FISH test. Photographs showing results from the PF-FISH test performed on thin blood smears in patients from Peru and Kenya infected with: (**A**) *P. falciparum*; (**B**) 1. *P. malariae* and 2. *P. falciparum*; (**C**) *P. ovale*; (**D**) *P. vivax*. The *P. falciparum*-specific probe and *Plasmodium* genus-specific probe fluoresce green and red, respectively, in the same field when viewed with appropriate light filters. The scale bars represent approximately 5 µm. Figure reproduced with permission under the creative commons license from Reference [39].

**Figure 3 diagnostics-12-01286-f003:**
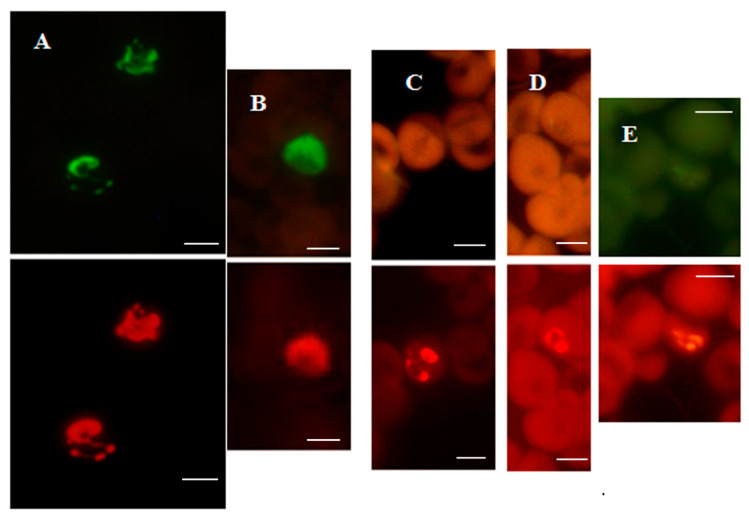
*P. vivax*-specific FISH test.Photographs showing PV-FISH test results on patient blood samples with independently confirmed malaria infections from Peru (**A**), India (**B**) and Kenya (**C**–**E**). Patient blood positive for (**A**,**B**) *P. vivax*; (**C**) *P. ovale*; (**D**) *P. malariae*; (**E**) *P. falciparum*. Green and red fluorescence are due to hybridization with the *P. vivax*-specific probe and *Plasmodium* genus-specific probe, respectively, in the same field when viewed with appropriate light filters. The scale bars represent approximately 5 µm. Figure reproduced with permission under the creative commons license from Reference [39].

**Figure 4 diagnostics-12-01286-f004:**
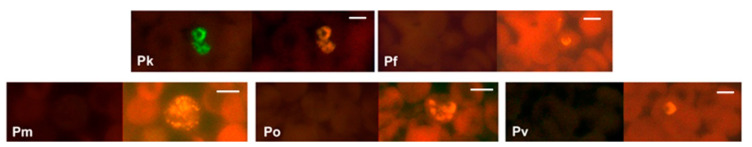
Specificity of the PK-FISH test for *P. knowlesi.* Photographs showing PK-FISH test results with the *P. knowlesi*-specific probe (green fluorescence) and the *Plasmodium* genus-specific probe (orange fluorescence) in blood smears containing *P. knowlesi* from monkey blood (**Pk**), and from human blood with confirmed infections of *P. falciparum* (**Pf**), *P. malariae* (**Pm**), *P. ovale* (**Po**) and *P. vivax* (**Pv**). Each set of paired photographs shows fluorescence in the same field when viewed in a LED fluorescence microscope with appropriate light filters (Appendix A). The scale bars represent approximately 5 µm. Reproduced with permission under the creative commons license from Reference [42].

**Figure 5 diagnostics-12-01286-f005:**
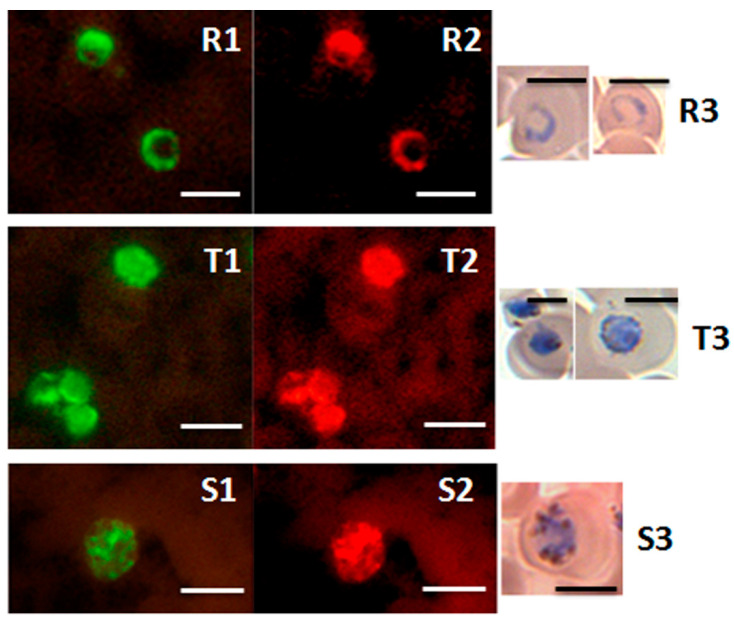
Detection of ring, trophozoite and schizont stages of *P. knowlesi* in the PK-FISH test. Photographs showing results from the PK-FISH test with the *P. knowlesi*-specific probe (green fluorescence) and the *Plasmodium* genus-specific probe (orange fluorescence) on R—rings; T—trophozoites; S—schizonts. Dual colour fluorescence in the same field is shown in paired photographs **R1** and **R2**, **T1** and **T2**, and **S1** and **S2**. Fluorescence was viewed in a LED fluorescence microscope with pertinent light filters (Appendix A)**.** The ring, trophozoite and schizont-stage parasites were produced from synchronised in vitro cultures of *P. knowlesi*. Parasites stained with Giemsa from smears prepared in parallel to the corresponding smears used in the PK-FISH test are shown in **R3**, **T3** and **S3** respectively. The scale bars represent approximately 5 µm. Reproduced with permission under the creative commons license from Reference [42].

**Figure 6 diagnostics-12-01286-f006:**
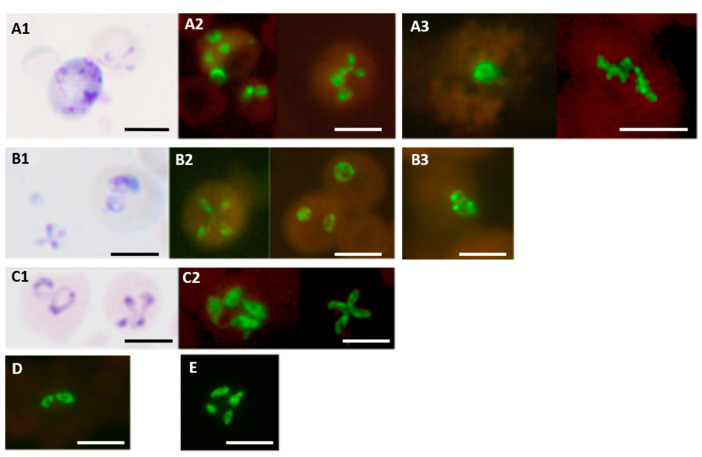
*Babesia* genus-specific FISH test on different *Babesia* species. Parasites stained with Giemsa in smears used for FISH tests are shown in the case of (**A1**) *B. microti*; (**B1**) *B. duncani*; and (**C1**) *B. divergens.* Fluorescence observed in FISH tests on corresponding smears from the same preparations are shown in (**A2**) *B. microti* (from hamster blood); (**B2**) *B. duncani* (from hamster blood); (**C2**) *B. divergens* (from culture). Other FISH test results on smears of (**D**) *B. bovis* (from bovine blood); (**E**) *B. bigemina* (from bovine blood) are also shown. Fluorescence in FISH tests on blood smears from two patients with the infecting species confirmed by DNA sequencing are shown in (**A3**) for *B. microti* and (**B3**) for *B. duncani*. Scale bars represent approximately 5 μm. Reproduced with permission under the creative commons license from Reference [43].

**Figure 7 diagnostics-12-01286-f007:**
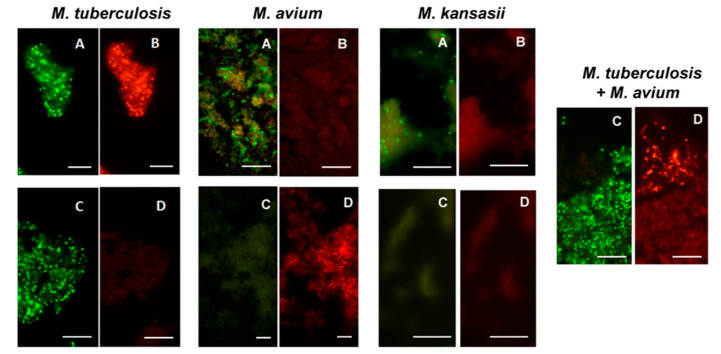
Dual color fluorescence reactivity of *Mycobacterium tuberculosis, Mycobacterium avium* and *Mycobacterium kansasii* in the MN Genus-MTBC FISH and MTBC-MAC FISH tests. Paired photographs showing dual colour fluorescence in the same microscopic field with **A**-MN Genus- specific probe (green fluorescence) and **B**-MTBC-specific probe (orange fluorescence) in the MN Genus-MTBC FISH test; and **C**-MTBC- specific probe (green fluorescence) and **D**-MAC-specific probe (orange fluorescence) in the MTBC-MAC FISH test. Mycobacteria used in the FISH tests were reference cultures of *M. tuberculosis*, *M. avium*, and *M. kansasii*, as well as an artificially mixed culture of *M. tuberculosis* and *M. avium*. Scale bars represent approximately 50 µm. Reproduced with permission under the creative commons license from Reference [28].

**Figure 8 diagnostics-12-01286-f008:**
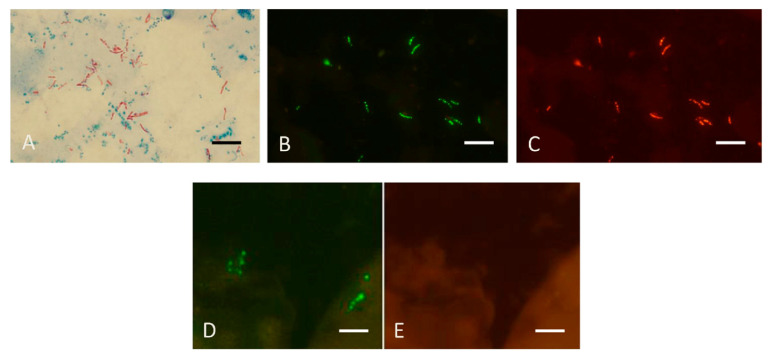
Sputum smears in the MN Genus-MTBC FISH test. Photographs showing dual colour fluorescence reactivity of sputum smears from patients with *Mycobacterium tuberculosis* and *Mycobacterium abscessus* infections in the MN Genus-MTBC FISH test. **A** shows Ziehl–Neelsen staining for acid-fast bacilli in sputum from a patient with *Mycobacterium tuberculosis* infection. MN Genus-MTBC FISH test results with the MN Genus- and MTBC-specific probes on the sputum of the same patient are shown in **B** (green fluorescence) and **C** (orange fluorescence), respectively. Reactions with the MN Genus- and MTBC-specific probes in the same field in a sputum smear from another patient with *Mycobacterium abscessus* infection are shown in **D** (green fluorescence) and **E** (orange fluorescence), respectively. Scale bars represent approximately 5 µm. Reproduced with permission under the creative commons license from Reference [29].

**Table 1 diagnostics-12-01286-t001:** FISH Tests for Identifying Pathogens.

Pathogen Type	Test Targets	References
**1. Bacteria**		
*Mycobacterium tuberculosis* complex (MTBC) & genus *Mycobacterium*	Cultures, biopsied tissue and sputum	[12,13,14,16,26,27,28,29]
*Mycobacterium avium complex* (MAC)	Cultures and biopsied tissue	[15,28,29,30]
*Mycobacterium leprae*	Skin and other biopsied tissue	[16,31]
*Gardnerella vaginalis* & *Lactobacillus* species	Cultures and tissue	[18,32]
*Bartonella* species including *B. berkhoffii*, *B. henselae*, *B. quintana*, and *B. vinsonii*	Blood smear	[33]
**2. Fungi**		
*Pneumocystis carinii*	Bronchoalveolar lavage and sputum	[11]
*Candida albicans*	Blood cultures	[17]
**3. Protozoa**		
*Cryptosporidium parvum*	Insect tissue *	[34]
*Giardia lamblia*	Insect tissue *	[34]
*Trypanosoma brucei gambiense* and related *Trypanozoons*	Blood and tissue smears	[35]
*Leishmania* species	Slit skin smear and formalin-fixed, paraffin-embedded tissues	[36,37]
*Trichomonas vaginalis*	Vaginal fluid	[38]
*Plasmodium falciparum*	Blood smear	[39]
*Plasmodium vivax*	Blood smear	[39]
*Plasmodium* genus	Blood smear	[39,40,41]
*Plasmodium knowlesi*	Blood smear	[42]
*Babesia* species	Blood smear and kidney tissues	[43,44,45,46]

* The same FISH test can potentially be applied to human tissues.

**Table 2 diagnostics-12-01286-t002:** Comparison of two NAA and FISH tests for malaria and tuberculosis.

	NAA Tests–PCR & LAMP	FISH
**Equipment & Facilities**	Comparatively expensive PCR machine with high maintenance cost. UV light source for LAMP. Clean room for all NAA.	Light microscope with LED/filter attachment (Appendix A); 37° incubator. Low maintenance cost.
**Test Cost & Reagent Stability**	$23–$28 per Xpert test [100] & similar for LAMP [101,102,103]. Refrigeration/freezing needed for reagents.	<$5 per test [42,95]. Reagents stable at 30 °C for several months [42].
**Personnel**	Highly trained operator for PCR and LAMP.	Trained microscopist.
**Test time**	<5 h for PCR & LAMP.	<2 h
**Throughput**	Automated for PCR. PCR more economical for large number of samples. LAMP usually read manually.	Individual samples and not presently automated. Amenable to automation by flow cytometry [24] and fluorescence detection by digital imaging.
**Laboratory and Location Suitability**	(i) *Malaria*: PCR rarely used for primary diagnosis except zoonotic malaria. LAMP rarely used for primary diagnosis of malaria. (ii) *Tuberculosis:* LAMP comparable to Xpert for tuberculosis [101]. Xpert not advantageous in locations with low levels of multi drug resistant (MDR) *M. tuberculosis* [101,102,103] or low disease prevalence [103]. LAMP not useful in areas with high levels of MDR [101].	All types of laboratories, locations and field use. Does not presently detect MDR *M. tuberculosis.*
**Species Identification**	(i) *Malaria*: Complex NAA methods can identify *Plasmodium* species [56]. (ii) *Tuberculosis*: Xpert only identifies MTB as do common LAMP tests.	(i) *Malaria*: FISH identifies *Plasmodium* genus and individual *Plasmodium* species [39,42]. (ii) *Tuberculosis*: FISH identifies MTBC and MAC in culture, sputum [28,29] and biopsied tissue [30].
**Limit of Detection**	(i) *Malaria*: <4 *Plasmodium*/ µL blood by PCR [55,56]. (ii) *Tuberculosis*: 1.3 × 10^2^ cfu/mL for *M. tuberculosis* in sputum with Xpert [104].	(i) *Malaria*: 55–84 *Plasmodium*/µL blood [39,42]. (ii) *Tuberculosis*: 2.2×10^4^ cfu/mL for MTBC in sputum [29].
**Specificity**	(i) *Malaria*: Up to 100% for common *Plasmodium* species by PCR [105], 98–99% by LAMP [106]. (ii) *Tuberculosis*: ≥96% for MTB with different Xpert models [107] & >90% with LAMP [108], in culture confirmed sputum in both cases.	(i) *Malaria*: For DNA-sequenced clinical samples >93.4% for important *Plasmodium* species [39,42]. (ii) *Tuberculosis*: 95.5% for DNA-sequenced MTBC in sputum in India and 100% for sputum derived cultures from India, Peru & USA [28,29].
**Sensitivity to Inhibitors in Clinical Samples**	PCR and LAMP sensitive to inhibitors in some tissue and sputum samples [83,96,97].	No FISH inhibitors in clinical samples.
**Detection of Live vs. Dead Pathogens**	PCR and LAMP detect DNA in both dead and live cells because of DNA stability [109]. Cell morphology remains unknown.	Detects live organisms only because rRNA degrades rapidly in dying cells [39,110]. Cell morphology visible. Useful for monitoring drug treatment & disease course

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
