# Peer review of "Fluorescence In Situ Hybridization (FISH) Tests for Identifying Protozoan and Bacterial Pathogens in Infectious Diseases"

_diagnostics, 2022, doi:10.3390/diagnostics12051286_

Round 1

Reviewer 1 Report

In this manuscript, the authors summarized the fluorescence in situ hybridization (FISH) assays used for the diagnosis of malaria, tuberculosis and babesiosis. Overall, the manuscript is well written and organized, but some improvements are necessary to emphasize the uniqueness of this review article. Therefore, I suggest the acceptance of this manuscript after accommodating the following comments.

1) The authors need to compare FISH assays with PCR-based tests and isothermal amplification-based assays in the table.

2) There might be bottlenecks to inhibit the application of FISH assays in resource-constrained diagnostic laboratories. The authors need to discuss the drawbacks/limitations of FISH methods. In addition, the future perspectives or direction of FISH assays should be provided.

3) The key requirements or considerations for FISH need to be discussed.

4) To make this manuscript more informative, the authors need to summarize the sequence of DNA or PNA probes (genus and species-specific sequences for Malaria, Tuberculosis, babesiosis) in table.

5) If there any specific reasons to choose three target diseases (malaria, babesiosis, and tuberculosis), the authors need to provide the rationale for choosing these three in this review article.

6) In line 311, Figure 1 should be corrected into Figure 8.

7) In Figure 1-4, there are no scale bar, which should be updated.

Reviewer 2 Report

In this article, authors have reviewed fluorescence in situ hybridization (FISH) tests used for diagnosing malaria, tuberculosis and babesiosis.  Identifying pathogens is important for diagnosing and then treating many infectious dis
eases and FISH has great potential for detecting  ribosomal 
RNA molecules in the cytoplasm of bacterial and protozoan pathogens. FISH is cheaper t and requires much less infrastructure relative  to the most standard approach based on PCR. This is a nice review article which emphasize the advantages of FISH for the detection of the pathogens in more complex and serious infectious diseases such as malaria, tuberculosis, and babesiosis. Overall the manuscript is written well, I believe the minor comments below could help to further enhance the visibility of this review article that attract a broad audience.  

  1. Since this is a review article, providing the details (merits and demerits) general (common) approaches for the detection of pathogens/bacterial microcolonies such as confocal microscopy, optical coherence tomography (OCT) [1-2] other than FISH would be benefical for the readers. Merits of OCT such as high-resolution 2D and 3D label free (unstained imaging) imaging of bacterial colonies and high resolution could be emphasized. Similarly, for confocal imaging, imaging stacks of bacteria / pathogens enabling enhanced visualization of with cellular level resolution could also be highlighted. ( https://doi.org/10.1117/1.JBO.21.12.127002, https://doi.org/10.1117/12.2190106)
  2. Figure 8 must be resized / enlarged to align with text. Unable to see any features in the current image settings used.
